# Targeted Therapy with PI3K, PARP, and WEE1 Inhibitors and Radiotherapy in HPV Positive and Negative Tonsillar Squamous Cell Carcinoma Cell Lines Reveals Synergy while Effects with APR-246 Are Limited

**DOI:** 10.3390/cancers15010093

**Published:** 2022-12-23

**Authors:** Karin Byskata, Monika Lukoseviciute, Filippo Tuti, Mark Zupancic, Ourania N. Kostopoulou, Stefan Holzhauser, Tina Dalianis

**Affiliations:** 1Department of Oncology-Pathology, Karolinska Institutet, Karolinska University Hospital, 171 64 Stockholm, Sweden; 2Department of Head-, Neck-, Lung- and Skin Cancer, Theme Cancer, Karolinska University Hospital, 171 64 Stockholm, Sweden

**Keywords:** targeted therapy, PI3K inhibitors, PARP inhibitors, WEE1 inhibitors, APR-246, HPV, tonsillar cancer, base of tongue cancer, oropharyngeal cancer, head and neck cancer

## Abstract

**Simple Summary:**

Human papillomavirus positive tonsillar and base of tongue cancers are rising in incidence but chemoradiotherapy has not improved survival, so novel treatments are needed. Therefore, targeted therapy with phosphoinositide 3-kinase, poly-ADP-ribose-polymerase, and WEE1 inhibitors was pursued alone or combined with/without radiotherapy or using APR-246 and their effects were analyzed by WST-1 viability, proliferation, and cytotoxicity assays on tonsillar and base of tongue cancer cell lines. All three inhibitors induced dose dependent inhibition of viability and proliferation and their effects were enhanced by adding irradiation or by combining the inhibitors. However, adding irradiation to already synergistic inhibitor combinations did generally not induce additional inhibitory effects. Only very high APR-246 doses decreased viability, so its use was limited in this context. To conclude, the effects of the inhibitors were frequently enhanced by adding radiotherapy or by combining them, but adding radiotherapy to the latter did not generally induce further inhibition.

**Abstract:**

Human papillomavirus positive (HPV^+^) tonsillar and base of tongue cancer (TSCC/BOTSCC) is rising in incidence, but chemoradiotherapy is not curative for all. Therefore, targeted therapy with PI3K (BYL719), PARP (BMN-673), and WEE1 (MK-1775) inhibitors alone or combined was pursued with or without 10 Gy and their effects were analyzed by viability, proliferation, and cytotoxicity assays on the TSCC/BOTSCC cell lines HPV^+^ UPCI-SCC-154 and HPV^−^ UT-SCC-60A. Effective single drug/10 Gy combinations were validated on additional TSCC lines. Finally, APR-246 was assessed on several TSCC/BOTSCC cell lines. BYL719, BMN-673, and MK-1775 treatments induced dose dependent responses in HPV^+^ UPCI-SCC-154 and HPV^−^ UT-SCC-60A and when combined with 10 Gy, synergistic effects were disclosed, as was also the case upon validation. Using BYL719/BMN-673, BYL719/MK-1775, or BMN-673/MK-1775 combinations on HPV^+^ UPCI-SCC-154 and HPV^−^ UT-SCC-60A also induced synergy compared to single drug administrations, but adding 10 Gy to these synergistic drug combinations had no further major effects. Low APR-246 concentrations had limited usefulness. To conclude, synergistic effects were disclosed when complementing single BYL719 BMN-673 and MK-1775 administrations with 10 Gy or when combining the inhibitors, while adding 10 Gy to the latter did not further enhance their already additive/synergistic effects. APR-246 was suboptimal in the present context.

## 1. Introduction

Tonsillar and base of tongue squamous cell carcinoma (TSCC/BOTSCC), the two major oropharyngeal squamous cell carcinomas subtypes, have a better outcome when they are human papillomavirus positive (HPV^+^) compared with when they are HPV-negative (HPV^−^) and the incidences of the former are increasing [1,2,3,4,5,6,7,8,9,10,11]. Current chemoradiotherapy of TSCC/BOTSCC has severe side effects, but this intensified treatment has not improved survival for the fraction of HPV^+^ TSCC/BOTSCC with poor outcome compared to previously given radiotherapy (RT), so new therapies are urgently needed [12,13,14,15]. 

Targeted therapy is an option that is promising and increasingly in focus for optimizing cancer treatments. More specifically, great progress has been made in triple negative breast cancer, where the use of specific inhibitors primarily targeting specific mutations has been introduced and even better effects are achieved by combining some of them [16,17]. Moreover, various approaches with a plethora of drugs and their combinations have been investigated, and these have disclosed many more promising effects [18,19,20,21,22,23,24,25]. To our knowledge, however, so far, very few of these current breakthroughs have been utilized for the treatment of HPV^+^ TSCC/BOTSCC, which is the focus of this pursuit. 

To better tailor HPV^+^ TSCC/BOTSCC therapy, many attempts have been made to identify prognostic and targetable markers of which most were initially defined by immunohistochemistry, but now are more often investigated by molecular or sequencing techniques [26,27,28,29,30,31,32,33,34,35,36,37,38]. We and others have, in HPV^+^ TSCC/BOTSCC, shown the frequent occurrence of mutations in the phosphatidyl-inositol-4,5-bisphosphate 3-kinase catalytic subunit alpha (*PIK3CA*) and fibroblast growth factor receptor 3 (*FGFR3*) genes [27,38]. Moreover, *PIK3CA* and *FGFR3* mutations or changes in their expression have sometimes been correlated to worse outcome, so targeting these genes may be of benefit, especially upon recurrence [27,39,40,41]. For breast cancer with *PIK3CA* mutations and urothelial cancer with *FGFR3* translocations/mutations, the Food and Drug Administration (FDA) has approved phosphoinositide 3-kinases (PI3K) and fibroblast growth factor receptor (FGFR) inhibitors, respectively, for clinical use [42,43]. 

Due to early inhibitor studies in vitro on breast and urothelial cancer, we also initially explored the effects of some of the then available non-FDA approved PI3K and FGFR inhibitors (BEZ235, BKM120 and AZD4547) on TSCC/BOTSCC cell lines HPV^+^ UM-SCC-47, UPCI-SCC-154, and HPV-UT-SCC-60A [44]. These lacked *PIK3CA/FGFR3* mutations, but still showed dose dependent responses and synergy [44]. We then validated these findings with FDA approved PI3K and FGFR inhibitors (BYL719 and JNJ-42756493 resp.) in TSCC cell lines HPV^+^ CU-OP-2, -3, -20, and HPV^−^ CU-OP-17 with/without (w/wo) *PIK3CA*/*FGFR3* mutations and could confirm dose dependent responses and synergy [45]. In addition, we assessed the effects of PI3K and Cyclin-Dependent-Kinase-4/6 (CDK4/6) inhibitor combinations on the HPV^+^/HPV^−^ CU-OP lines and revealed positive synergistic effects similar to that for certain breast cancer types [17,46]. 

More recently, we also tested single and combined poly-ADP-ribose-polymerase (PARP) and WEE1 inhibitors on the CU-OP lines [46,47]. However, here, despite dose dependent responses with single drug administrations, synergy was less frequent compared to the synergistic effects of these drugs earlier reported for triple negative breast cancer with cyclin E or BRCA1 alterations and cancer of the biliary tract [16,17,48]. 

Clearly, some, but not all above combinations, could be useful clinically for relapsed HPV^+^ TSCC/BOTSCC, similar to that for other cancers, especially upon failure or resistance to anti PD-1/PDL-1 therapy, often given today, but before doing so, more knowledge would be of value [16,17,44,45,46,47,48,49]. For this reason and due to earlier positive data by others and us using targeted therapy [16,17,44,45,46,47,48,49], we wanted to extend these approaches by testing prior and novel combinations and cell lines. 

In this study, we explored the effects of PI3K, PARP, and WEE1 inhibitors (BYL719, BMN-673, MK-1775, respectively) alone and the recent BMN-673 and MK-1775 combination [46] as well as additional (BYL719 and BMN-673 and BYL719 and MK-1775) combinations on two well-established cell lines, HPV^+^ UPCI-SCC-154 and HPV^−^ UT-SCC-60A. More importantly, in parallel, on the same two cell lines, we investigated whether adding irradiation (IR) to the aforementioned inhibitors alone or their combinations could improve their efficacy, since IR has been shown to enhance the effect of PARP inhibitors in other systems [50]. We then employed some of the best above combinations on two CU-OP cell lines. Finally, since MK-1775, known to target mutated p53, was previously shown to be efficient as a single drug in decreasing viability and proliferation of the CU-OP cell lines [46], we also wanted to explore whether APR-246 targeting TP53 and sometimes restoring its function could be useful in our system [51]. We therefore investigated the effects of APR-246 on all of the above indicated HPV^+^/HPV^−^ TSCC/BOTSCC cell lines. 

## 2. Materials and Methods

### 2.1. Cell Lines and Seeding

UPCI-SCC-154, a tongue squamous cell carcinoma with wt *TP53* and HPV^−^ UT-SCC-60A, a TSCC with mutant *TP53,* respectively, were obtained from Susan Gollin, University of Pittsburgh USA and Reidar Grénman, University of Turku, Finland, respectively, and both were kept in Dulbecco’s modified Eagle’s medium (Gibco) [52,53]. HPV^+^ CU-OP-2 (with a *PIK3CA* and *FGFR3* mutation), CU-OP-3, CU-OP-20 (with a *PIK3CA* mutation), and HPV^−^ CU-OP-17, all with wt *TP53*, were obtained from N. Powell, Cardiff University UK [47,54]. They were kept in Glasgow minimum Essential medium (GMEM) (Merk Life Science UK, Limited, Dorset, UK) on 60 Gy irradiated 3T3 fibroblasts as feeder layers as described earlier [47,54]. All media were complemented with 10% fetal bovine serum (FBS; Gibco), 1% L-glutamine, 100 U/mL of penicillin, and 100 µg/mL streptomycin, and all cells were maintained at 37 °C in a humidified incubator with 5% CO_2_.

For analysis, UPCI-SCC-154 and UT-SCC-60A were seeded with 5000 cells/well, while the CU-OP lines (without feeders) were seeded at 7500 cells/well with all in 96-well plates in 90–200 μL media. All experiments were repeated at least three times.

### 2.2. Drugs, IR, and Treatments

FDA approved phosphoinositide 3-kinase (PI3K) inhibitor BYL719, poly-ADP-ribose-polymerase (PARP) inhibitor BMN-673 as well as non-FDA approved WEE1 inhibitor MK-1775 were all purchased at Selleckhem Chemicals Munich, Germany. Stock solutions at 10 mM for all inhibitors were diluted in the presence of 1% DMSO. Further dilutions were, however, all made with PBS and stored at −20 °C. The inhibitors were diluted once again before use (i.e., 24 h after cell seeding) and utilized in the following doses: BYL719 0.5–20 μM; PD-0332991 5–40 μM; and for BMN-673 0.1–50 μM and MK-1775 0.1–50 μM. APR-246 was kindly obtained from Prof. Klas Wiman, Karolinska Institutet, Stockholm, Sweden and stocked in DMSO and then further diluted 10–100 μM in PBS before use. 

IR with 10 Gray (Gy), was given with an X-Rad 225XL machine (PXi Precision X-ray, North Brandford, CT, USA) 3 hours (h) before inhibitor treatment to UPCI-SCC-154 and UT-SCC-60A, while to CU-OP cells, 10 Gy was given at the same time or 3 h after inhibitor treatment, if not stated otherwise.

### 2.3. WST-1 Viability Assay

Viability pursued for 72 h after treatment was assessed by a WST-1 assay (Roche Diagnostics, Mannheim, Germany) in accordance with the instructions of the manufacturer and repeated three times, as previously reported in more detail [44].

### 2.4. Cell Proliferation and Cytotoxicity Assays

The 96-well plates with the various cell lines were placed into the IncuCyte S3 Live® Cell Analysis System utilizing the Incucyte™ Cytotox Red Reagent (Essen Bioscience, Welwyn Garden City, UK) and followed for proliferation (cell confluence) and cytotoxicity by taking images every 2 h, as described earlier [45]. Of the three repeated experiments, a representative one is shown.

### 2.5. Statistical Analysis

To verify the efficacy of the single inhibitors or their combinations in comparison to the negative PBS control, a multiple t-test followed by correction for multiple comparison of the means conferring to the Holm Sidak method was performed [44,45,46,55,56]. The combined effects based on at least three separate viability assays were evaluated by applying the effect-based approach ‘Highest Single Agent’ [55,56], the details of which have previously been presented [44,45,46,55,56]. 

## 3. Results

### 3.1. Viability of HPV^+^ UPCI-SCC-154 and HPV^−^ UT-SCC-60A Lines after Exposure to PI3K, PARP and WEE1 Inhibitors BYL719, BMN-673 and MK-1775 Alone or Combined with IR and of HPV^+^ CU-OP-2 and 20 to BYL719 wo IR Measured by WST-1 Assays

Previously, we have shown that HPV^+^ CU-OP-2, -3, and -20 and HPV^−^ CU-OP-17 presented dose dependent responses to BYL719, BMN-673, and MK-1775 [46]. Here, we first tested single low dose effects of 0.5–1 µM of the three above inhibitors or 10 Gy alone, the latter based on previous experience [57], and then combined these three inhibitors with 10 Gy on UPCI-SCC-154 and UT-SCC-60A, two well-established TSCC/BOTSCC cell lines. Upon assaying the inhibition of viability, we found dose dependent responses to single drug administrations, with UPCI-SCC-154 generally being slightly more sensitive to all three drugs compared to UT-SCC-60A (Figure 1A–L). We then evaluated the effects of adding 10 Gy to the inhibitors.

#### 3.1.1. BYL719 with/without IR

For UPCI-SCC-154, without IR, 0.5 and 1.0 µM BYL719 significantly decreased viability compared to PBS at all timepoints except with 1.0 µM after 24 h (at least *p* < 0.05) (Figure 1A). For UT-SCC-60A without IR, this was the case only with the 1.0 µM dose at 24 and 48 h (at least *p* < 0.05) but not after 72 h (Figure 1C). Upon adding 10 Gy to BYL719, a significant reduction in viability was seen in both cell lines with all doses and timepoints except with 0.5 µM after 24 h (for all others at least *p* < 0.05) (Figure 1B,D). 

#### 3.1.2. BMN-673 with/without IR

Both UPCI-SCC-154 and UT-SCC-60A showed dose dependent responses to BMN-673 without IR (Figure 1E,G). A significant decrease in viability of UPCI-SCC-154 was noted with 0.5 and 1.0 µM BMN-673 after 48 h and for the higher dose also after 72 h (all at least *p* < 0.05) (Figure 1E). Upon combination with 10 Gy, stronger responses were observed with 0.5 µM at 72 h and with 1.0 µM at all timepoints (all at least *p* < 0.05) (Figure 1F). UT-SCC-60A was less sensitive than UPCI-SCC-154 to BMN-673 alone, especially with the 0.5 µM dose, with a significant decrease in viability only with 1.0 µM BMN-673 after 72 h (at least *p* < 0.05) when IR was not applied (Figure 1G). Upon combining BMN-673 with 10 Gy, the decrease in viability was not improved at any of the timepoints in comparison to administering 10 Gy alone (Figure 1H). 

#### 3.1.3. MK-1775 with/without IR

UPCI-SCC-154 was fairly sensitive to MK-1775, which also induced a strong decrease in viability after 24–72 h with lower doses both w/wo IR, except for 1.0 µM MK-1775 without IR at 24 h (at least *p* < 0.05) (Figure 1I,J). For UT-SCC-60A, 1.0 µM MK-1775, even without IR, decreased viability at all timepoints, however, when adding 10 Gy to the lower dose, an enhanced significant decrease in viability at 24 and 72 h was observed (at least *p* < 0.05) (Figure 1K,L).

#### 3.1.4. Combination Indices with the “Highest Single Agent” Approach

Combinational indices (CIs) 48 and 72 h after treatment of BYL719, BMN-673, and MK-1775, w/wo IR, were calculated for UPCI-SCC-154 and UT-SCC-60A to determine the effect of combinations (Figure 2).

In line with Figure 1, for UPCI-SCC-154, adding 10 Gy to BYL719 and BMN-673 resulted in positive combinational effects with CIs < 1 at both 48 and 72 h, while neutral effects were observed upon combining MK-1775 with IR, the latter likely due to the high sensitivity of this cell line to MK-1775 (Figure 2A–C). 

For UT-SCC-60A, again in line with the viability data, clear positive combinational effects were observed at 48 h for the BYL719 and 10 Gy and MK-1775 and 10 Gy combinations, while the 10 Gy and BMN-673 combinations mainly resulted in neutral or adverse effects (Figure 2D–F). 

To conclude, for UPCI-SCC-154, adding 10 Gy to BYL719 and BMN-673 enhanced the efficacy of single drugs, while for the 10 Gy and MK-1775 inhibitor combination, neutral effects were obtained. For UT-SCC-60A, enhanced efficacy was shown with the 10 Gy BYL719 and 10 Gy MK-1775 combinations, but not by combining 10 Gy with BMN-673. 

#### 3.1.5. Validation of the Single BYL719, BMN-673, and MK-1775 Drug Treatments with/without RT in CU-OP-2 and -20 with WST-1 Viability Assays and by Calculating the Combinational Indices 

Due to the positive effects above of single drugs and IR, we decided to conduct some complementary experiments on two additional cell lines. A pilot study on CU-OP-2, with 10 Gy given at the same time or 3 h after treatment with 0.5 µM BYL719 showed mainly positive effects, irrespective of the timing of IR (data not shown). 

In all of the following experiments with CU-OP-2 and -20, and 0.5 and 1 µM BYL719 or BMN-673 or MK-1775 were given alone, or with 10 Gy at the same time. Single low doses of BYL719 or BMN-673 showed hardly any effects for CU-OP-2 and marginal effects on CU-OP-20, while single doses of MK-1775 were consistently slightly more efficient and showed a significant difference for CU-OP-2 48 h with 0.5–1 µM doses and 0.5 µM after 24 h for CU-OP-20 (for all at least *p* < 0.05) (Appendix A). Moreover, when adding 10 Gy to the single inhibitors, a significantly enhanced decrease in viability was shown in CU-OP-2 after 24 h (for all at least *p* < 0.05), while for CU-OP-20, this increase was not so pronounced (Appendix A). When calculating the combinational indices at 48 and 72 h, a similar pattern to what was disclosed in the viability assays was observed with CI < 1 for CU-OP-2 for all three drug and 10 Gy combinations, while for CU-OP-20, this was mainly not the case (Appendix A).

#### 3.1.6. Summary of BYL719, BMN-673 and MK-1775 Single Inhibitors with/without IR

To summarize, UPCI-SCC-154 was generally more sensitive to single doses of the inhibitors than UT-SCC-60A, however, both cell lines were most sensitive to MK-1775 and more resistant to BMN-673 with the drug doses used here. 

Adding IR with 10 Gy frequently decreased the viability in comparison to using single drugs alone in both cell lines, especially 48–72 h after treatment. This was illustrated in detail by calculating the combinational effects [55,56], showing the benefits of combining BYL719 and IR and BMN-673 and IR for UPCI-SCC-154, and BYL719 and IR and MK-1775 and IR for UT-SCC-60A. Furthermore, positive additive/synergistic effects with BYL719, BMN-673, and MK-1775 and IR were also observed for CU-OP-2, the more resistant line, while this was not as much the case for CU-OP-20, the more sensitive cell line.

### 3.2. Viability of HPV^+^ UPCI-SCC-154 and HPV^−^ UT-SCC-60A Lines upon Exposure to Combinations of PI3K, PARP, and WEE1 Inhibitors BYL719, BMN-673, and MK-1775 Respectively, with/without IR Measured by WST-1 Assays

Previously, HPV^+^ CU-OP-2, -3, and -20 and HPV^−^ CU-OP-17 were shown to present synergistic responses to BMN-673 combined with MK-1775, but not as often and also to a lesser extent when compared to the BYL719 and CDK4/6 combination [46]. Here, we tested additional cell lines (i.e. HPV^+^ UPCI-SCC-154 and HPV^−^ UT-SCC-60A) and more combinations, in other words, 0.5–1 µM each of either BYL719 and MK-1775, BYL719, and BMN-673 and the recently examined combination BMN-673 and MK-1775, and now w/wo IR with 10 Gy (Figure 3).

#### 3.2.1. BYL719 and MK-1775 with/without IR

All combinations of BYL719 with MK-1775 wo IR decreased the viability significantly compared to PBS in both cell lines (at least *p* < 0.05) (Figure 3A,C). Upon adding 10 Gy, significant declines in viability were observed at all timepoints for all BYL719 and MK-1775 combinations, except for 0.5 µM BYL719 with 1.0 µM MK-1775 after 72 h in UPCI-SCC-154, and after 24 h in UT-SCC-60A (at least *p* < 0.05) (Figure 3B,D). 

#### 3.2.2. BYL719 and BMN-673 with/without IR

All BYL719 with BMN-673 combinations, both w/wo IR, decreased viability after 48 and 72 h in UPCI-SCC-154, except at 72 h for the 1.0 µM BYL719 and 0.5 µM BMN-673 and IR combination (at least *p* < 0.05) (Figure 3E,F). In UT-SCC-60A, all BYL719 and 1.0 µM BMN-673 combinations decreased the viability at all timepoints (at least *p* < 0.05) and this was enhanced further upon applying 10 Gy, since significance was reached at all timepoints except for combinations with 0.5 µM BYL719 at 24 h (at least *p* < 0.05) (Figure 3G,H). 

#### 3.2.3. BMN-673 and MK-1775 with/without IR

For UPCI-SCC-154, all BMN-673 with MK-1775 combinations decreased the viability significantly after 48 and 72 h, except for the 0.5 µM BMN-673 and 1.0 µM MK-1775 combination at 48 h (at least *p* < 0.05) (Figure 3I). In UT-SCC-60A, most BMN-673 and MK-1775 combinations, except for all BMN-673 combinations with 0.5 µM MK-1775 at 24 h, provided a significant decrease in viability (at least *p* < 0.05) (Figure 3K). Combining 10 Gy with BMN-673 and MK-1775 showed a statistically significant strong inhibition at all timepoints for both UPCI-SCC-154 and UT-SCC-60A at most timepoints (*p* < 0.05) (Figure 3J,L).

#### 3.2.4. Combination Indices with the “Highest Single Agent” Approach 

Combinational indices (CIs) of BYL719, BMN-673, and MK-1775, w/wo IR, were calculated for each cell line 48 and 72 h after treatment to determine the effect of the various combinations, and these showed mainly positive or neutral effects (Figure 4). 

UPCI-SCC-154 is often more sensitive to single drug treatments than UT-SCC-60A, so we hypothesized that CIs would thereby indicate more positive effects for the latter compared to the former cell line (Figure 4) due to previous experience [44,45,46]. Adding IR to the inhibitor combinations used here did not generally enhance the already efficient inhibition obtained by combining two inhibitors (Figure 4).

*BYL719/MK-1775 combinations* resulted exclusively in positive effects in UT-SCC-60A, while for UPCI-SCC-154, the effects were more neutral. Adding 10 Gy did not change the effects at 48 h but at 72 h, giving IR was likely less positive than wo IR for both cell lines. 

*BYL719/BMN-673 combinations* showed positive effects in both cell lines w/wo IR.

*BMN-673/MK-1775 combinations* showed positive effects in both cell lines with lower doses of MK-1775, while higher MK-1775 doses provided either a more neutral or even negative combinational effect and with positive effects being greater in UT-SCC-60A compared to UPCI-SCC-154. Adding 10 Gy had marginal effects in both cell lines.

#### 3.2.5. Summary of BYL719, BMN-673 and MK-1775 Combination Treatments w/wo IR 

To summarize, BYL719/MK-1775, BYL719/BMN-673, or BMN-673/MK-1775 combinations were generally quite efficient and superior to using one inhibitor alone, with UPCI-SCC-154 generally being more sensitive to most combined treatments compared to UT-SCC-60A. However, adding 10 Gy to already efficient combinations did not usually improve the inhibition of viability further, and although IR tended to possibly boost the effect of all inhibitor combinations after 48 h (Figure 3), this was not always evident when calculating the CIs (Figure 4). 

### 3.3. Viability of HPV^+^ UPCI-SCC-154, CU-OP-2, -3 and -20 and HPV^−^ UT-SCC-60A and CU-OP-17 Cell Lines to APR-246 Measured by WST-1 Assays

As already mentioned, APR-246 has been shown to restore TP53 function in some cases [42]. Due to the previous positive effects of MK-1775 also targeting TP53 on the CU-OP cell lines [36] and now on UPCI-SCC-154 and UT-SCC-60A, we tested the effects of 10–70 µM APR-246 on HPV^+^ UPCI-SCC-154, CU-OP-2, -3, -20, and HPV^−^ UT-SCC-60A and CU-OP-17 (Figure 5).

All cell lines were relatively resistant to APR-246. Only the highest dose (70 µM) was able to decrease the viability significantly after 72 h for HPV^+^ UPCI-SCC-154, CU-OP-2, CU-OP-20, and HPV^−^ CU-OP-17 (at least *p* < 0.05), while this was still not the case for HPV^+^ CU-OP-3 and HPV^−^ UT-SCC-60A (Figure 5). These findings were relatively discouraging and not pursued further with additional assays.

### 3.4. Proliferation and Cytotoxicity Responses of HPV^+^ UPCI-SCC-154 and HPV^−^ UT-SCC-60A Lines after Exposure to PI3K, PARP, and WEE1 Inhibitors BYL719, BMN-673, and MK-1775 alone w/wo IR and Proliferation of HPV^+^ CU-OP-2 and -20 Treated with BYL719 and MK-1775 alone or w/wo IR

Due to the positive effects with a decrease in viability using single administrations of 0.5 µM or 1.0 µM of either BYL719, BMN-673 and MK-1775 together with 10 Gy, we here examined the effects of these treatments and corresponding doses on cell confluence and cytotoxicity on HPV^+^ UPCI-SCC-154 and HPV^−^ UT-SCC-60A. To validate the data, with some very efficient combinations in the viability tests, the same doses of BYL719 and MK-1775 w/wo 10 Gy were used to follow their effects on the proliferation of the HPV^+^ CU-OP-2 and -20 cell lines. 

#### 3.4.1. Proliferation Responses of HPV^+^ UPCI-SCC-154 and HPV^−^ UT-SCC-60A Lines after Exposure to PI3K, PARP, and WEE1 Inhibitors BYL719, BMN-673, and MK-1775 alone w/wo IR

Cell confluence was followed for 72 h after treatment with 0.5 µM or 1.0 µM of either BYL719, BMN-673, and MK-1775 alone or together with 10 Gy on the UPCI-SCC-154 and HPV^−^ UT-SCC-60A lines. Compared to PBS, low doses of BYL719 and BMN-673 had marginal effects on the cell confluence on both cell lines, while the effects of MK-1775 were slightly more pronounced, but when adding 10 Gy to the single inhibitors, a prominent decease was observed in cell confluence, especially for the UT-SCC-60A cell line with BYL719 and MK-1775, but not BMN-673, in line with the viability data (Figure 6).

#### 3.4.2. Proliferation Responses of HPV^+^ CU-OP-2 and CU-OP-20 Lines after Exposure to PI3K, PARP, and WEE1 Inhibitors BYL719 and MK-1775 alone w/wo IR

To also complement the viability data, we followed the effects on proliferation of single inhibitors BYL719 and MK-1775 (the two most efficient drugs) with the same doses used as in the viability tests for HPV^+^ CU-OP-2 and -20. Similar to the viability data, we observed marked effects on cell confluence when using BYL719 and MK-1775 together with 10 Gy, especially for CU-OP-2 and BYL719 (Appendix A). Cytotoxicity was not followed up in these cell lines.

#### 3.4.3. Cytotoxic Responses of HPV^+^ UPCI-SCC-154 and HPV^−^ UT-SCC-60A Lines after Exposure to PI3K, PARP, and WEE1 Inhibitors BYL719, BMN-673, and MK-1775 alone w/wo IR

In parallel with cell confluence, the cytotoxic effect was followed for 72 h after treatment with 0.5 µM or 1.0 µM of either BYL719, BMN-673, and MK-1775 w/wo 10 Gy on UPCI-SCC-154 and HPV^−^ UT-SCC-60A. Compared to PBS, low doses of the inhibitors had virtually no or marginal effects on cytotoxicity in either cell line, however, adding 10 Gy enhanced the effect of MK-1775, but not for BYL719 and BMN-673 on both cell lines (Figure 7). 

#### 3.4.4. Summary of BYL719, BMN-673 and MK-1775 Single Treatments w/wo IR

To summarize, single BYL719, BMN-673, or MK-1775 treatments showed marginal effects on the confluence and cytotoxicity when used alone on HPV^+^ UPCI-SCC-154 and HPV^−^ UT-SCC-60A, and this was also the case for the effects on the confluence for CU-OP-2 and CU-OP-20 with BYL719 and MK-1775. Adding 10 Gy usually improved the inhibition of cell confluence on all of the tested cell lines, while the effects of adding IR were marginal on cytotoxicity when tested on HPV^+^ UPCI-SCC-154 and HPV^−^ UT-SCC-60A, with the exception of some enhancement of the cytotoxic effects of MK-1775. 

### 3.5. Proliferation and Cytotoxicity Responses of HPV^+^ UPCI-SCC-154 and HPV^−^ UT-SCC-60A Lines after Combined Treatments with PI3K, PARP, and WEE1 Inhibitors (BYL719, BMN-673 and MK-1775, Respectively) w/wo IR 

Due to the additive/synergistic of BYL719, BMN-673, and MK-1775 on HPV^+^ UPCI-SCC-154 and HPV- UT-SCC-60A by viability, we investigated their corresponding effects using the same doses used for the viability tests on proliferation and cytotoxicity.

#### 3.5.1. Proliferation Responses of HPV^+^ UPCI-SCC-154 and HPV^−^ UT-SCC-60A Lines after Exposure to PI3K, PARP, and WEE1 Inhibitors BYL719, BMN-673, and MK-1775 in Different Combinations w/wo IR

Cell confluence was followed for 72 h after treatment with 0.5 µM or 1.0 µM of either BYL719, BMN-673, and MK-1775 in different combinations w/wo 10 Gy on the UPCI-SCC-154 and HPV^−^ UT-SCC-60A lines. Compared to PBS, combinations of BYL719/BMN-673 BYL719/MK-1775, and BMN-673/MK1775 all had some effect on the cell confluence on both cell lines, while the effects of adding 10 Gy to the inhibitor combinations only provided marginal additional efficacy on cell confluence (Figure 8).

#### 3.5.2. Cytotoxic Responses of HPV^+^ UPCI-SCC-154 and HPV- UT-SCC-60A Lines after Exposure to PI3K, PARP, and WEE1 Inhibitors BYL719, BMN-673, and MK-1775 in Different Combinations w/wo IR

In parallel with cell confluence, the cytotoxic effect was followed for 72 h after treatment with 0.5 µM or 1.0 µM of combinations of BYL719, BMN-673, and MK-1775 w/wo 10 Gy on UPCI-SCC-154 and HPV^−^ UT-SCC-60A. Compared to PBS, low doses of all of the combined inhibitors w/wo IR had some effect on UPCI-SCC-154, and this was also the case for UT-SCC-60A, except for the BYL719/BMN-673 combination, however, adding IR did not increase the cytotoxic effect to any major extent in this context (Figure 9). 

#### 3.5.3. Summary of BYL719, BMN-673 and MK-1775 Single Treatments w/wo IR 

To summarize, combined BYL719, BMN-673, or MK-1775 treatments showed improved effects on the decrease of confluence and there were some effects on the cytotoxicity in most cases, with the exception of the BYL719/BMN-673 combination on UT-SCC-60A. When adding 10 Gy, the already enhanced inhibition of cell confluence did not improve immensely, which was mainly the case for the effects on cytotoxicity.

## 4. Discussion

In this study, we examined the effects of PI3K, PARP, and WEE1 inhibitors (BYL719, BMN-673, MK-1775, respectively) alone or in different combinations w/wo IR (10 Gy) on two well-established TSCC/BOTSCC cell lines HPV^+^ UPCI-SCC-154 and HPV^−^ UT-SCC-60A. We then validated some of the most favorable combinations on two TSCC lines, HPV^+^ CU-OP-2 and -20, the former considered relatively resistant and the other as a sensitive cell line [44,45,54]. Finally, due to the sensitivity of the cell lines to MK-1775 targeting TP53, we also explored the effects of APR-246 (another drug targeting TP53) on several HPV^+^/HPV^−^ TSCC/BOTSCC cell lines.

Dose dependent responses to single BYL719, BMN-673, and MK-1775 treatments were found in HPV^+^ UPCI-SCC-154 and HPV^−^ UT-SCC-60A, with the former being generally more sensitive than the latter and by adding 10 Gy, mainly positive or synergistic effects were obtained. Similar data were seen with BYL719, BMN-673, and MK-1775 w/wo 10 Gy on CU-OP-2, while these effects were not at all pronounced for the more sensitive cell line CU-OP-20. 

Combining BYL719/BMN-673, BYL719/MK-1775, or BMN-673/MK-1775 in HPV^+^ UPCI-SCC-154 and HPV^−^ UT-SCC-60A generally decreased the viability much more than using one drug alone in both cell lines, however, adding 10 Gy did not markedly improve this already efficient decrease in the viability in any of the cell lines. 

Finally, only relatively high concentrations of APR-246 exhibited a prominent decrease on all of the tested cell lines. 

The dose dependent responses found in UPCI-SCC-154 and UT-SCC-60A with single BYL719, BMN-673, and MK-1775 administrations was expected, since this has been demonstrated before with corresponding inhibitors on other tumor cell lines by others and us [16,45,46,48]. Moreover, while the responses were dose dependent, the sensitivity of the various cell lines did vary in line with previous data on additional cell lines [46]. However, not many studies have reported the use of combining these single inhibitors with IR, and to our knowledge, this is one of the few studies to have done so with the TSCC/BOTSCC cell lines. 

Adding 10 Gy to BYL719 posed additive effects on HPV^+^ UPCI-SCC-154 and CU-OP-2 as well as HPV^−^ UT-SCC-60A, but not so much on CU-OP-20, irrespective of whether they had *PIK3CA* mutations or not. We presently have no direct explanation for this positive finding, but PI3K inhibitors have been shown to interact with IR in other systems [50]. For example, combining pan-PIK inhibitors or combining PI3Kα inhibitors with IR have been shown to enhance the inhibition of proliferation of lung cancer cell lines in vitro and in vivo compared to using either approach alone [58]. Likewise, we have also previously described that IR can enhance the effect of the PI3K inhibitor BYL719 in medulloblastoma cell lines [59]. Consequently, this approach may have a beneficial effect for several cancers including TSCC/BOTSCC.

Combining 10 Gy with BMN-673 showed different patterns. While HPV^+^ UPCI-SCC-154 showed positive responses to that combination, this was not the case for UT-SCC-60A, despite the two cell lines being relatively equally sensitive to 10 Gy. Similarly, HPV^+^ CU-OP-2 and CU-OP-20 showed differing responses, with the former being sensitive and the latter insensitive to the BMN-673 10 Gy combination.

The fact that both UPCI-SCC-154 and UT-SCC-60A were relatively insensitive to treatment with BMN alone is, however, not surprising. We have previously shown that the TSCC/BOTSCC cell lines, independent of HPV status, can be either sensitive or resistant to the PARP inhibitor olaparib [47]. Furthermore, we have recently disclosed that all CU-OP cell lines used here were relatively insensitive to low doses (0.1 and 0.5 µM) of BMN-673 [46]. 

In contrast, in other systems, it has been shown that PARP inhibitors, together with IR, have very good effects such as in BRCA1(+/−) lymphoblastoid cells, hepatocellular cancer and lung and breast cancer xenografts [60,61,62]. Moreover, in a triple negative breast cancer phase I trial, the authors found that treatment was well-tolerated and that this approach should be investigated further in additional trials [63]. However, due to the variable sensitivity of TSCC/BOTSCC lines to PARP inhibitors [46] and the present report, combining PARP and IR on TSCC/BOTSCC may not be a first choice for all, despite some promising effects that have been presented for PARP, cetuximab, and RT/IR (for review see [64]). In this review, the identification of head and neck cancers, apart from those that have *BRCA-1* and *2* mutations, which would be sensitive to PARP inhibitors by using different biomarkers, was pointed out as an important issue and discussed in some detail [64]. Clearly, applying this combination could be efficient in some cases, but more information regarding the timing of treatment and for which type of tumors needs to acquired. 

Notably, both HPV^+^ UPCI-SCC-154 and HPV^−^ UT-SCC-60A were responsive to low doses of MK-1775, similar to what we have shown before (and here) for the CU-OP-cell lines and in line with data by others on the HPV^+^/HPV^−^ head and neck cancer cell lines [46,65]. Furthermore, UT-SCC-60A demonstrated a positive response to MK-1775 and 10 Gy. We hypothesize that because UT-SCC-60A has a *TP53* mutation, it is more sensitive to MK-1775 than to BYL719 and BMN-673, while instead, UPCI-SCC-154 with no *TP53* mutation and being HPV^+^ also exhibited positive effects to PARP inhibitors, as previously shown for other HPV^+^ cell lines [46] (for review see [64]). 

Combining WEE1 inhibitors with IR has positive effects, as has been shown before for other cancer types such as leukemic T-cells, osteosarcoma, or cells with TP53 mutations such as lung and breast cancer cell lines [66,67,68]. To our knowledge, this specific combination (WEE1 and IR) has not been tested extensively specifically for TSCC/BOTSCC cell lines, despite the existence of a few reports, but in those cases, WEE1 has been included in other types of combinations in head and neck cancer [65,69,70]. Comparing our data to similar or somewhat different investigations, some interesting observations were noted. One report focused on the synergistic effects of using AZD-1775 (corresponding to MK-1775) and cisplatin, and effects on apoptosis were observed in the HPV^+^ head and neck cancer cell lines, but there was no investigation with regard to RT/IR [65]. In another study, with focus on radiosensitization using Chk1 inhibitors, it was shown that combining Chk1 and WEE-1 inhibitors resulted in very efficient radiosensitization toward the HPV^+^ cell lines [69]. In a more recent report, the effects of Chk1, PARP, and WEE1 inhibitors on radiosensitization were investigated in vitro and in vivo [70]. In that report, the authors suggested that modest preferential radiosensitization could be observed by WEE1 and PARP2 inhibitors in the HPV^−^ cell lines, whereas Chk1 inhibitors were more efficient in HPV^+^ tumor models [70]. 

Clearly, there are some data using various inhibitors in combination with IR, however, these are not always concordant, suggesting that these issues need to be further clarified. It is most likely more complicated, since the mutational profiles of the individual tumors vary, and harboring, for example, *TP53* mutations or various *BRCA-1* or *2* mutations or others, may affect the different responses to the various combinations, irrespective of the HPV status of the tumors. It is likely that identifying HPV status alone will be insufficient and one will have to characterize the tumors in more detail before applying specific individualized therapies [64,69,70]. 

Combining BYL719/BMN-673, BYL719/MK-1775, and BMN-673/MK-1775 mainly elicited positive effects on both cell lines and decreased the viability with low doses. 

This was of note, but not entirely unexpected, although the latter combination (BMN-673/MK-1775) in our hands had not previously shown the most optimal synergy on the CU-OP cell lines [46]. Obviously, much may depend on the nature of the specific cell lines. Nevertheless, it has been suggested by others that combining PARP inhibitors that induce DNA damage with WEE1 inhibitors and Ataxia telangiectasia mutated (ATR) inhibitors should be of use and of interest [64]. Moreover, the former combination is currently under investigation in the OLAPCO study (NCT02576444), a phase II signal-searching study [56]. In the latter study, one anticipates that it will be possible to identify which patients may respond to such treatments (NCT02576444) [64]. Obviously, even here, there could be more individualistic responses.

The BYL719 and BMN-673 combination seemed useful in both cell lines in this study. This would then be in line with other studies, where PARP inhibitors have been combined with PI3K inhibitors for high grade serous ovarian and breast cancer with some antitumor effects [71,72]. The authors concluded that these drugs could be used together, but that one would need to decrease the PI3K inhibitor (BKM120) dose [72]. Moreover, in a phase II study, whether adding a PI3K inhibitor to a PARP inhibitor compared to using a PARP inhibitor alone was suggested to be further investigated [72].

The final inhibitor combination in this report was using PI3K together with a WEE1 inhibitor. Here, an additive/synergistic effect was also mainly observed on UT-SCC-60A, most likely due to it being the most resistant cell line, while for UPCI-SCC-154, already sensitive to MK-1775, the additive/synergistic effect was not as pronounced. We attempted to compare our data to similar reports, but were not able to find larger studies using this approach in head and neck cancer, instead, both were often combined separately (e.g. with PARP inhibitors) [71,72,73,74]. 

Adding IR to the three BYL719/BMN-673, BYL719/MK-1775, and BMN-673/MK-1775 beneficial combinations did not really improve their effects further in a pronounced way. However, in some cases, an earlier response and more marked decrease in viability was observed already after 48 h with regard to the UT-SCC-60A cell line, but this was not markedly reflected when calculating the CIs. 

Nevertheless, it was of interest to follow the effects of single inhibitors alone w/wo IR as well as to follow the effects of combined inhibitors w/wo IR. The data suggest that depending on the tumor and the situation, one could design a treatment that would include a specific inhibitor together with IR or use two inhibitors in combination. For these, further studies would be of interest, however, the decisive choice would then be either upon recurrence, if the patient tumor does not respond to the possibilities used today, or in the future for primary treatment. However, for the latter, further studies are of importance.

Finally, in this report, we also explored the effect of APR-246 on quite a few cell lines with the anticipation that it would possibly show some effects because high risk HPV E6 downregulates TP53 or because UT-SCC-60A has a documented TP53 mutation [75,76]. However, none of the cell lines exhibited a marked sensitivity to APR-246, indicating that this drug is most likely not the drug of choice in this context. One could also argue that the *TP53* mutations in tumors sensitive to WEE1 are not similar to the ones where APR-246 has its best effects, since the UT-SCC-60A line has a p.Arg342Ter (c.1024C > T), which initiates a premature stop codon in exon 9 of *TP53* and results in a truncated TP53 protein [77,78,79]. It has been suggested that APR-246 may exert its best efficacy in cell lines that have TP53 mutations primarily in exons 5–8 (where the majority are located) such as in breast cancer or in cancers where it has a non-functional conformation [80,81]. Furthermore, the authors also suggest that one should combine many of the drugs targeting TP53 with regular cytostatic drugs for better efficacy [81].

There were limitations in this study, in that we had a small number of inhibitors and cell lines. However, we did validate some of the most important combinations, and it is obvious that the variability between the different cell lines, irrespective of HPV status, poses additional issues. Obviously, it is not possible to generalize on the sensitivity of the tumors to specific inhibitors or their combinations w/wo IR, solely by HPV status. In the future, a profound molecular analysis of the various cancers should be conducted before the selection of treatment, and experiments carried out to follow if the selection was optimal.

In summary, combining 10 Gy with single BYL719 BMN-673 and MK1775 inhibitors can potentially be of use for TSCC/BOTSCC, but is likely not needed when already combining inhibitors exhibiting synergy, while APR-246 was not optimal in this context. Nevertheless, without more detailed analysis of the individual tumors, it may not be possible to predict which combination is the best. 

## 5. Conclusions

In this study, single PI3K, PARP, and WEE1 inhibitors BYL719, BMN-673, and MK-1775 induced dose dependent inhibition of viability and proliferation in the HPV^+^/HPV^−^ TSCC/ BOTSCC cell lines, and when the inhibitors were combined with 10 Gy, or used in combination, mainly positive/synergistic effects were obtained. However, adding 10 Gy to the already efficient postive/synergistic inhibitor combinations had no major further effects. Furthermore, low APR-246 concentrations had limited effects. 

To conclude, adding 10 Gy to single BYL719, BMN-673, and MK-1775 low dose administrations or combining these inhibitors notably enhanced their efficacy, while adding 10 Gy to the latter did not induce any major further improvement of their already enhanced effects. APR-246 was suboptimal in this context. 

## Figures and Tables

**Figure 1 cancers-15-00093-f001:**
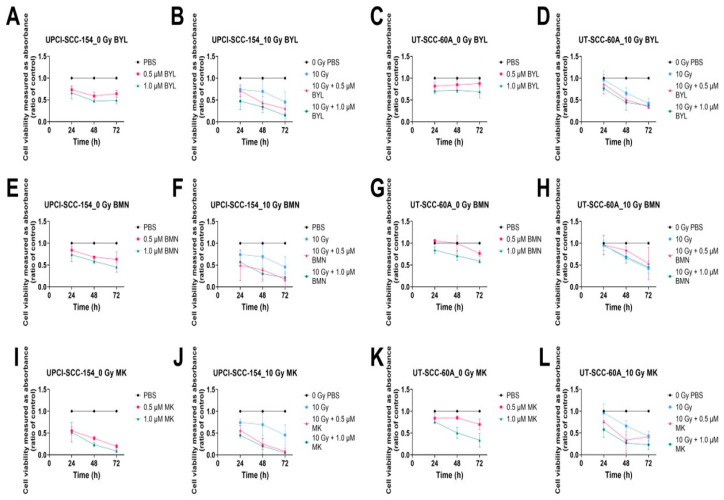
WST-1 viability assays on HPV^+^ UPCI-SCC-154 and HPV^−^ UT-SCC-60A with viability measured as absorbance 24–72 h after single treatments with 0.5–1 µM BYL719 (**A**–**D**), BMN-673 (**E**–**H**), and MK-1775 (**I**–**L**), with/without 10 Gy. BYL denotes BYL719, BMN denotes BMN-673, and MK denotes MK-1775.

**Figure 2 cancers-15-00093-f002:**
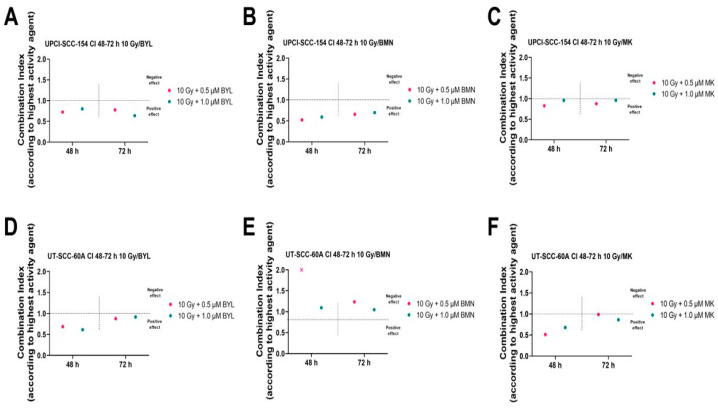
Combinational effects on HPV^+^ UPCI-SCC-154 and HPV^−^ UT-SCC-60A upon adding 10 Gy to PI3K inhibitor BYL719 (**A**,**D**), to PARP inhibitor BMN-673 (**B**,**E**), and to WEE-1 inhibitor MK-1775 (**C**,**F**). Combination indices (CIs) were shown with the highest single agent approach after 48 and 72 h, where CI < 1 shows a positive and CI > 1 a negative combination effect. CIs were calculated from the mean of three experiments analyzed by WST-1 assays at 48 and 72 h after treatment. BYL denotes BYL719, BMN denotes BMN-673, and MK denotes MK-1775.

**Figure 3 cancers-15-00093-f003:**
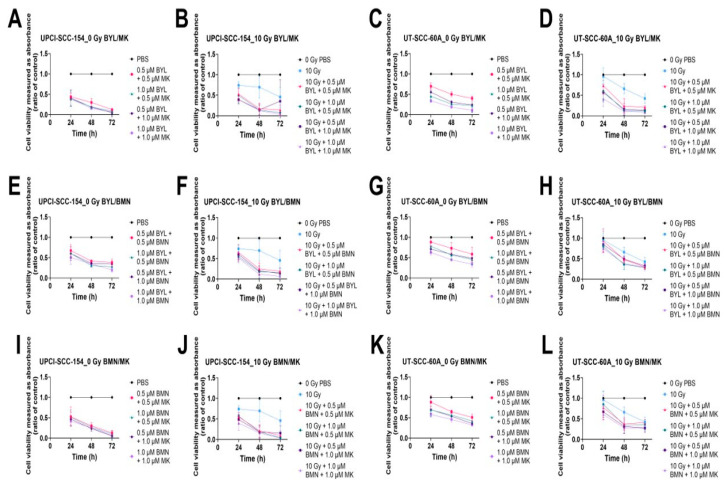
WST-1 assay on the HPV^+^ UPCI-SCC-154 and HPV^−^ UT-SCC-60A lines. Cell viability measured as absorbance after 24, 48, and 72 h of combined treatments with BYL719/MK-1775 (**A**–**D**), BYL719/BMN-673 (**E**–**H**), and BMN-673/MK-1775 (**I**–**L**) with/without 10 Gy. BYL denotes BYL719, BMN denotes BMN-673, and MK denotes MK-1775.

**Figure 4 cancers-15-00093-f004:**
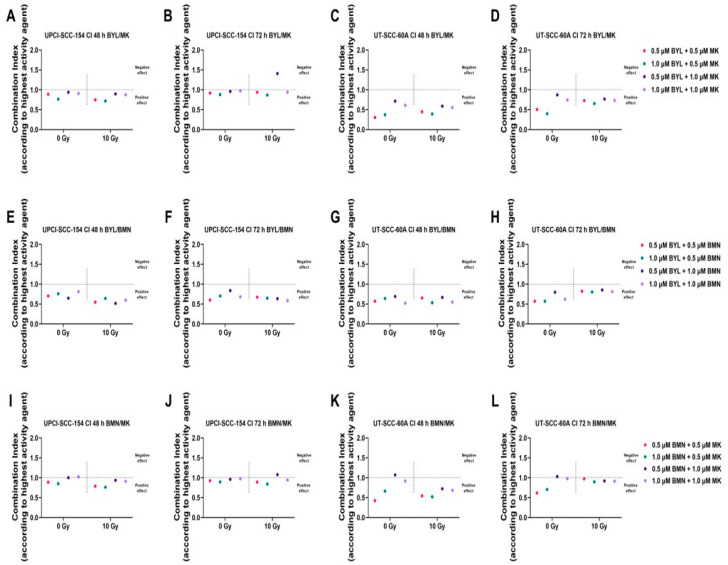
Combination indices (CIs) of BYL719/MK-1775 (**A**–**D**), BYL719/BMN-673 (**E**–**H**), and BMN-673/MK-1775 (**I**–**L**) with 0 or 10 Gy on HPV^+^ UPCI-SCC-154 and HPV^−^ UT-SCC-60A cell lines after 48 and 72 h of treatment. CIs were shown with the highest single agent approach after 48 and 72 h, where CI < 1 shows a positive and CI > 1 a negative combination effect. BYL denotes BYL719, BMN denotes BMN-673, and MK denotes MK-1775.

**Figure 5 cancers-15-00093-f005:**
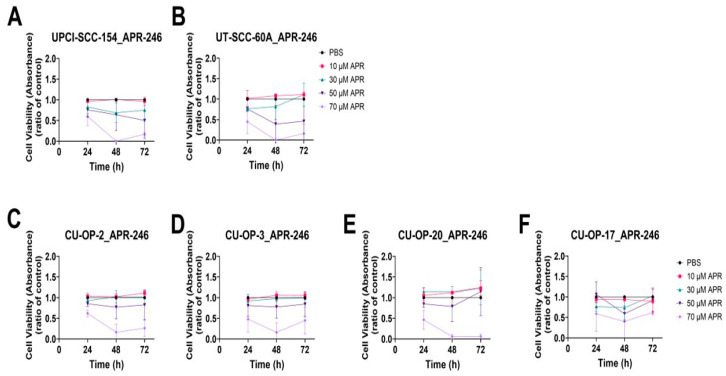
WST-1 viability assay on the HPV^+^ UPCI-SCC-154 and HPV- UT-SCC-60A (**A**,**B**) and CU-OP-2, -3, -20 and -17 cell lines (**C**–**F**). Cell viability measured as absorbance after 24, 48, and 72 h of single treatments with APR-246. APR denotes APR-246.

**Figure 6 cancers-15-00093-f006:**
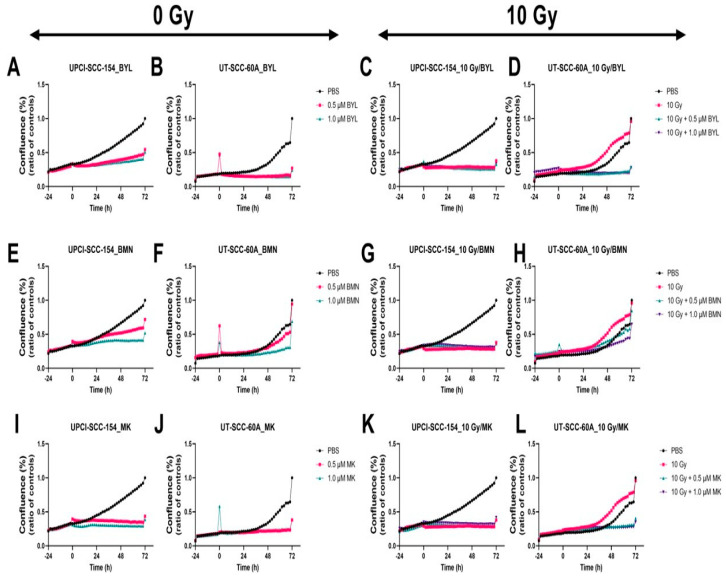
Proliferation responses of the HPV+ UPCI-SCC-154 and HPV− UT-SCC-60A cell lines upon treatment with BYL719 with/without (w/wo) 10 Gy (**A**–**D**), BMN-673 w/wo 10 Gy (**E**–**H**), or MK-1775 w/wo 10 Gy (**I**–**L**). The graphs represent one experimental run per cell line. Confluence (%) denotes the proliferation response, BYL denotes BYL719, BMN denotes BMN-673, and MK denotes MK-1775.

**Figure 7 cancers-15-00093-f007:**
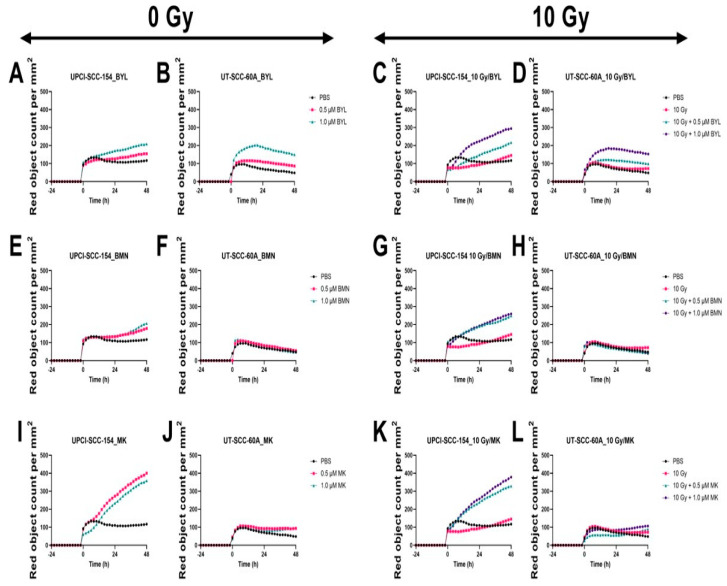
Cytotoxicity response of the HPV+ UPCI-SCC-154 and HPV− UT-SCC-60A cell lines upon treatment with BYL719 with/without (w/wo) 10 Gy (**A**–**D**), BMN-673 w/wo 10 Gy (**E**–**H**), or MK-1775 w/wo 10 Gy (**I**–**L**). The graphs represent one experimental run per cell line. BYL denotes BYL719, BMN denotes BMN-673, and MK denotes MK-1775.

**Figure 8 cancers-15-00093-f008:**
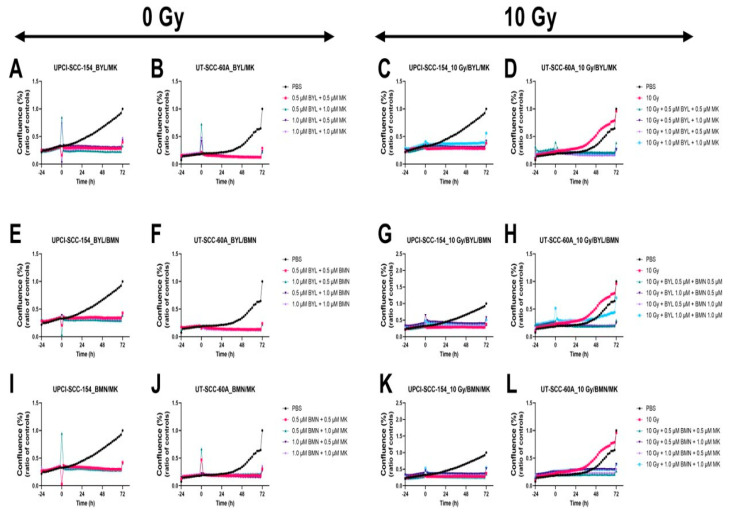
Proliferation responses of the HPV+ UPCI-SCC-154 and HPV− UT-SCC-60A cell lines upon treatment with MK-1775/BYL719 with/without (w/wo) 10 Gy (**A**–**D**), BYL719/BMN-673 w/wo 10 Gy (**E**–**H**), or MK-1775/BMN-673 w/wo 10 Gy (**I**–**L**). The graphs represent one experimental run per cell line. Confluence (%) denotes the proliferation response, BYL denotes BYL719, BMN denotes BMN-673, and MK denotes MK-1775.

**Figure 9 cancers-15-00093-f009:**
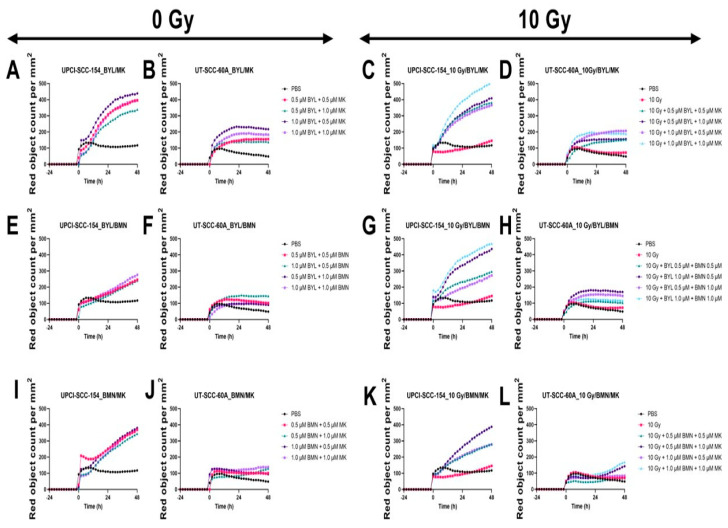
Cytotoxicity response of the HPV+ UPCI-SCC-154 and HPV− UT-SCC-60A cell lines upon treatment with MK-1775/BYL719 with/without (w/wo) 10 Gy (**A**–**D**), BYL719/BMN-673 w/wo 10 Gy (**E**–**H**), or MK-1775/BMN-673 w/wo 10 Gy (**I**–**L**). The graphs represent one experimental run per cell line. BYL denotes BYL719, BMN denotes BMN-673, and MK denotes MK-1775.

## Data Availability

The data presented in this study are available on request from the corresponding author.

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
