# Peer review of "Targeted Therapy with PI3K, PARP, and WEE1 Inhibitors and Radiotherapy in HPV Positive and Negative Tonsillar Squamous Cell Carcinoma Cell Lines Reveals Synergy while Effects with APR-246 Are Limited"

_cancers, 2022, doi:10.3390/cancers15010093_

Round 1
Reviewer 1 Report
Comments and Suggestions for Authors The article by Byskata at al. is devoted to an important topic. It will be interesting for a broad community of scientists, I recommend this paper to be published in the journal. Here are some minor suggestions:
1) The abstract is weak and meaningless. Authors should thoroughly improve it because it's the first impression.
2) The “Title”, “Keywords”, and “Discussion” of the manuscript should be more concise.
3) The language is ok, though an additional check would be beneficial.
4) More recent research information and references on “combination drug therapies” (Transl. Oncol. 2022, 16, 101332; Biomedicines. 2021, 9, 689; Cancers. 2018, 10, 483; Front. Pharmacol. 2021, 12, 712995) and “targeted therapy” (BMC Med. 2022, 20, 90; New J. Chem. 2022, 46, 17673-17677; Biomedicines. 2021, 9, 1591; Front. Pharmacol. 2021, 12, 778973.) should be added in “Introduction” to highlight the novelty of this work clearly. This is critical to address in this manuscript, the authors should enrich this part in the revised version for improved readability and wide readership.
5) To include abbreviations at the end of the manuscript - but I leave this to the authors' discretion.
Author Response
We thank you for your comments and advice and have modified our manuscript accordingly point by point below.
The article by Byskata at al. is devoted to an important topic. It will be interesting for a broad community of scientists, I recommend this paper to be published in the journal. Here are some minor suggestions:
1) The abstract is weak and meaningless. Authors should thoroughly improve it because it's the first impression.
We thank the reviewer for this comment and have improved both the simple summary and the abstract.
2) The “Title”, “Keywords”, and “Discussion” of the manuscript should be more concise.
-The Title of the manuscript has now been modified
-The Keywords have been modified and re-arranged in order to be more concise.
-The Discussion has been modified and shortened slightly to be more concise.
3) The language is ok, though an additional check would be beneficial.
-The English has been checked by a colleague with English as first language.
4) More recent research information and references on “combination drug therapies” (Transl. Oncol. 2022, 16, 101332; Biomedicines. 2021, 9, 689; Cancers. 2018, 10, 483; Front. Pharmacol.2021, 12, 712995) and “targeted therapy” (BMC Med. 2022, 20, 90; New J. Chem. 2022, 46, 17673-17677; Biomedicines. 2021, 9, 1591; Front. Pharmacol. 2021, 12, 778973.) should be added in “Introduction” to highlight the novelty of this work clearly. This is critical to address in this manuscript, the authors should enrich this part in the revised version for improved readability and wide readership.
We thank the reviewer for the information of these interesting publications, which indeed cover a broader field and have inserted these references as well as two provided by reviewer 2 into the introduction, which has been extended with a paragraph to briefly mention the development of the field targeted therapy and novel drugs.
5) To include abbreviations at the end of the manuscript - but I leave this to the authors' discretion.
We thank the reviewer for this comment, but since to our knowledge this is not usually done, we have not pursued this issue further.
Reviewer 2 Report
This review report has been removed from the review record as it did not meet MDPI’s review report standards (https://www.mdpi.com/reviewers#_bookmark11).
Reviewer 3 Report
Comments and Suggestions for Authors
So far, I cannot conclude that the methodology allowed the authors' conclusions.
1. Which cell lines are TSCC? Or BOTSCC? and which of them HPV+ and HPV-
2. There was no HPV+ effect, but the authors wanted to find need treatment ideas for HPV+ cancer
1. Simple abstract: "but chemoradiotherapy has not improved" - misleading, this is not the only therapy option, there is surgery, radiotherapy alone, immunotherapy.
2. Simple abstract: "but chemoradiotherapy has not improved" - misleading, as the the focus on improving tx compared to HPV- is to de-escalate tx
3. Simple abstract: "tonsillar and base of tongue cancer cell lines"- missing: HPV+ cell lines?
4. See 3: if HPV+: same effect in HPV- cell lines: where is the relation to HPV?
5. Simple abstract. "Only very high APR-246 ... was limited" - how do you know? Just cell line experiment
6. Abstract: "but chemoradiotherapy is not curative for all" - see above, and statement is wrong: is curative in may cases!
7. Abstract: "TSCC/BOTSCC cell lines" - meas what? TSCC or BOTSCC cell line? or a mix? Same for HPV- UT-SCC-60A.
8. Abstract: "additional TSCC lines" which lines are meant?
9. Abstract: Reading the results: The measured effects have what to do with HPV? The tx was dose-dependent effective in HPV+ and HPV- cell lines. Abstract starts with the need for better tx in HPV+ cancer
10. Introduction: "Current chemoradiotherapy of TSCC/BOTSCC has severe side effects" - see above, misleading. Many patients are treated with surgery, radiotherapy alone or neoadjuvant tx
11. Introduction: "reat progress has been made in triple negative breast
cancer," - no similarities to head and neck cancer? Why to mention here?
12. Introduction: "the focus of this pursuit" - but the Results do NOT show a HPV+ effect!
13. Introduction: "approved phosphoinositide 3-kinases (PI3K) and fibroblast growth factor receptor (FGFR) inhibitors"- PI3K inhibition already tested in head neck cancer: Oral Oncol
2022 Aug:131:105939. doi: 10.1016/j.oraloncology.2022.10593
14. Introduction: I miss a clear statement what goes really here beyond previous experiments (reference 46):_ Kostopoulou ON, Zupancic M, Pont M, Papin E, Lukoseviciute M, Mikelarena BA, Holzhauser S, Dalianis T.
Viruses. 2022 Jun 23;14(7):1372. doi: 10.3390/v14071372.
15. Introduction: "we also wanted to explore whether APR-246 targeting TP53" just in the last lines of the Introduction, APR-246 is introduced. This is not sufficient, rationale not explored like the others.
16. Methods: UPCI-SCC-154, a tongue squamous cell carcinoma -. so this is NOT TSCC/BOTSCC! So NOT of interest!
17. Methods: "HPV+ CU-OP-2(with a PIK3CA and FGFR3 mutation), CU-OP-3, CU-OP-20 (with a PIK3CA mutation),and HPVô€€€ CU-OP-17, " - are what? TSCC/BOTSCC?
18. Statistics: "multiple t-test" - only allowed when data are normally distributed. Tests for this are missing
19. Results: "Previously, we have shown" - is NOT Results - but Discussion
20. Figures: Statistics missing
21. Results 3.1Statistics missing
22. Results - ("at least p < 0.05") - in general, not acceptable to describe statistics. Give concrete numbers and data, and cohen's d effect size
23. Discussion: There is no HPV effect, not explored.